# Genome-Wide Identification, Expression Analysis, and Subcellular Localization of *Carthamus tinctorius* bHLH Transcription Factors

**DOI:** 10.3390/ijms20123044

**Published:** 2019-06-21

**Authors:** Yingqi Hong, Naveed Ahmad, Yuanyuan Tian, Jianyu Liu, Liyan Wang, Gang Wang, Xiuming Liu, Yuanyuan Dong, Fawei Wang, Weican Liu, Xiaowei Li, Xu Zhao, Na Yao, Haiyan Li

**Affiliations:** 1Ministry of Education Engineering Research Center of Bioreactor and Pharmaceutical Development, Jilin Agricultural University, Changchun 130118, China; hyq985676490@163.com (Y.H.); naveedjlau@gmail.com (N.A.); yuanzitian@163.com (Y.T.); ljy20180436@163.com (J.L.); wly19940427@163.com (L.W.); 13842847523@163.com (G.W.); xiuming1211@163.com (X.L.); yydong@aliyun.com (Y.D.); 13596076186@163.com (F.W.); liuweican602@163.com (W.L.); xiaoweili1206@163.com (X.L.); 2College of Life Sciences, Jilin Agricultural University, Changchun 130118, China; 3Jilin Institute for Food Control, Changchun 130033, China; plnto@163.com

**Keywords:** bHLH family, *Carthamus tinctorius*, gene expression, flavonoid biosynthesis, subcellular localization

## Abstract

The basic helix–loop–helix (bHLH) family is the second largest superfamily of transcription factors that belongs to all three eukaryotic kingdoms. The key function of this superfamily is the regulation of growth and developmental mechanisms in plants. However, the bHLH gene family in *Carthamus tinctorius* has not yet been studied. Here, we identified 41 bHLH genes in *Carthamus tinctorius* that were classified into 23 subgroups. Further, we conducted a phylogenetic analysis and identified 10 conserved protein motifs found in the safflower bHLH family. We comprehensively analyzed a group of bHLH genes that could be associated with flavonoid biosynthesis in safflower by gene expression analysis, gene ontology annotation, protein interaction network prediction, subcellular localization of the candidate CtbHLH40 gene, and real-time quantitative expression analysis. This study provides genome-wide identification of the genes related to biochemical and physiological processes in safflower.

## 1. Introduction

Transcription factors (TFs) are substantial key elements of various physiological and biochemical pathways in higher plant development. Regulation of genes related to growth, stress response of plants, and other developmental processes is mainly regulated by TF activation and repression [1,2]. The second largest TF family across eukaryotic kingdoms is the basic helix–loop–helix (bHLH) superfamily. Based on the highly conserved bHLH domain, the bHLH superfamily is categorized into two main groups: the basic region and the HLH region [3,4]. The basic region is a DNA-binding interface located next to the N-terminus of the protein domain, whereas the HLH region serves as the dimerization domain [1] and consists of two amphipathic α-helices linked by a loop. Apart from the high frequency of bHLH conserved domains, some other motifs also occur within the bHLH superfamily [5,6]. Based on the highly conserved domains and their evolutionary relationships, this superfamily is mainly divided into 20–25 subfamilies in agronomically important plants [6,7]. A wide range of extensive studies have been reported on the bHLH superfamily-related genes in plant species after genome sequences have been produced, including *Arabidopsis* [8], peanut [9], blueberry [10], tomato [11], Chinese cabbage [12], and apple [13]. Additionally, functional and structural characterizations of most plant bHLH proteins have been described in detail. These findings demonstrated that bHLH proteins are involved in various biochemical and physiological networks across the plant kingdom. Studies on bHLH revealed its partial involvement in iron uptake [14], response against salt and drought stress [13,15], tanshinone biosynthesis [16], and petal growth development. The most prominent studies on the bHLH family demonstrated its association with anthocyanin biosynthesis and regulation in flowering plants and fruits [17].

*Carthamus tinctorius* L. is the most extensively used flowering plant of family *Asteraceae* and is known for its medicinal value in West Asia and China. It is a rich resource of conjugated linoleic acid (CLA) and an essential herbal medicine with less side effects [18]. Safflower oil has been reported to minimize the risk of cardiovascular diseases and regulate cholesterol [19,20]. The biologically active ingredients of safflower petals are attributed to flavonoids that harbor diverse pharmacological effects for cancer, inflammation, hypertension, prevention of excessive oxidative stress, and blood flow regulation [21,22]. Of the 5000 types of plant flavonoids that have been discovered in a variety of plants to date, safflower shares a striking variety of important secondary metabolites, including quinochalcone glycoside, hydroxysafflor yellow A, safflor yellow B, and chalcone glycosides [23,24].

Safflower has a diploid genome that has not yet been sequenced; therefore, there is a lack of information on and understanding of the molecular mechanisms, functional diversity, and structural genomics in this important plant. Therefore, it is necessary to distinguish functional genes, gene expression factors, TFs, promoters, and chromosomal allocations. Our previous study [25] revealed a set of candidate genes in safflower that are potentially involved in the flavonoid metabolism. There are currently no comprehensive reports available on the safflower bHLH superfamily genes. In the present study, we aimed to investigate the comprehensive genome wide identification of the *Carthamus tinctorius* bHLH transcription factors family followed by spatiotemporal expression analysis and GPF tagging of the candidate genes. The objectives of this work were to: (1) extensively characterize the physicochemical properties of 41 safflower bHLH genes; (2) conduct a phylogenetic analysis of these genes and identify bHLH protein motifs; (3) conduct gene expression analysis of the selected 41 bHLH genes at different flowering stages and in different tissues; (4) produce expression profiles of the bHLH gene family using quantitative real-time PCR (qRT-PCR) analyses at various developmental stages; and (5) infer subcellular localization of a candidate gene, CmbHLH40, via GFP fusion protein assay. Our study provides an extensive genome-wide investigation and expression analysis of the safflower bHLH superfamily, which might have a crucial role in functional characterization of TFs involved in flavonoid metabolism.

## 2. Results and Discussion

### 2.1. Genome-Wide Identification and Phylogenetic Analysis of Safflower bHLH Genes

To identify bHLH Genes in safflower, an in silico search from safflower genome was performed based on keywords (*Carthamus tinctorius* L.) and a Pfam ID search of safflower genome database, identification of homologous *Arabidopsis* bHLH amino acid sequence, and a Pfam domain search of the conserved domain database (CDD). Based on the information available on genome sequencing, total of 89 safflower bHLH genes were obtained by sequencing. Some of them had short sequences and no full-length sequence and some have long sequences. After eliminating these invalid genes, the remaining genes were compared by BLAST, and the genes with very low homology ratio were removed. Some of them did not have the typical domain of bHLH. Finally, 41 genes were obtained. These 41 genes were divided into 15 groups (groups 1–15) (Table 1) based on conserved domains and sequence similarity based on phylogenetic analysis. The largest group was group 9, which contained seven members, whereas the smallest groups were groups 3, 5, 6, 12, 13, and 15, with only one member.

The physical and chemical parameters of most proteins in the same group were similar according to an analysis using the ProtParam online tool. The 41 CtbHLH-encoded amino acids ranged from 186–706; molecular weights ranged from 21.19 kDa (CtbHLH37) to 77.13 kDa (CtbHLH32), with an average of 43.38 kDa. Theisoelectric pointspIvalues ranged from 5.21 (CtbHLH16) to 9.34 (CtbHLH7), with an average of 6.64. A grand average of hydropathicity (GRAVY) index analysis showed that most of the CtbHLH proteins were hydrophilic. The most stable protein was CtbHLH18, which had a stability index of 31.23.

### 2.2. Phylogenetic Analysis

To subdivide the *Carthamus tinctorius* bHLH proteins into subfamilies and elucidate the evolutionary relationships between the *Carthamus tinctorius* bHLH proteins and those of *Arabidopsis*, multiple bioinformatics tools were used to construct a phylogenetic tree that consisted of 138 *Arabidopsis* bHLH proteins and 41 bHLH *Carthamus tinctorius* proteins (Figure 1). The aforementioned bHLH proteins clustered into 18 subfamilies, which is consistent with the tree topology and previous classification of the bHLH taxonomic group in genus *Arabidopsis* [3]. The 18 subfamilies were designated 1, 2, 3, 4, 6, 7, 9, 10, 11, 12, 13, 14, 16, 17, 19, 20, 22, and 23 in Figure 1. These subfamilies were consistent with those recently reported in some other plant species, including peach [26], tomato [11], Chinese cabbage [12], peanut [9], and wheat [27]. These findings revealed that bHLH proteins within the reported subfamilies that are present in dissimilar plant species could play a basic role in plant development and evolution. In addition, our findings were found consistent with the previous subfamily classification of the bHLH domain in other plants, where 15–25 subfamilies were reported [6]. However, taxonomic classification of the subfamilies has varied. For example, one group renamed the bHLH subfamilies using the English alphabet [4,13], and another group named them using Arabic numerals [3,28] however, most previous reports used the same pattern as Roman numerals [6,9,10,11,12,29]. Therefore, to classify the *Carthamus tinctorius* bHLH subfamilies, we adopted the Roman numeral-based taxonomic names in our analysis, following [3] and [6]. None of the *Carthamus tinctorius* bHLH proteins were sorted into subfamilies 5, 8, 15, 18, and 21, which indicated loss of those proteins throughout *Carthamus tinctorius* evolution. Overall, these results revealed that the absence of the non-conserved bHLH subfamilies in *Carthamus tinctorius* could have explicitly evolved to enhance organic processes of the plants or improve stress tolerance.

### 2.3. Motifs and Multiple Sequence Alignment of bHLH Proteins

To combine the phylogenetic tree parameters and thus the selection of the same taxonomic members, a tree was created using the 41 *Carthamus tinctorius* bHLH protein sequences (Figure 2). Although the order of the germline subfamily is slightly different from the order of the tree presented above (Figure 1), each member of the subfamily shared the same composition and pattern as found in the neighbor-joining tree of *Carthamus tinctorius* bHLH proteins. The conserved protein motifs of bHLH proteins were screened using the online tool, MEME version 4.8.1, by uploading 41 protein sequences of the *Carthamus tinctorius* bHLH superfamily according to [30]. Approximately all sequences were found to exhibit two types of highly conserved protein motifs with a similar width of 38 amino acids, which are demonstrated as red and green blocks, respectively (Figure 2). Furthermore, the locations of the motifs were investigated, and the majority of these motifs were present at the C-terminus of *Carthamus tinctorius* bHLH subfamilies 1, 3, 6, 8, and 14. Subfamily 9 shows the presence of some motifs adjacent to the N-terminus. In most of the cases, excluding a few bHLH proteins in safflower, these two conserved domains are located next to each other; however, some bHLH protein motifs were identified that were located far from each other. The maximum distance between the two conserved motifs was found in CtbHLH9. These findings are consistent with the previous findings on the bHLH domains in tomato [11] and wheat [27]. In this study, motif1 was indicated by red blocks and determined to be composed from basic residues and loop helix 1. However, motif2 was represented with green blocks and is composed of a loop and helix number 2. The variable loop length determines the gap between motif1 and motif2 in the bHLH protein superfamily. Motif 1 (in red) and motif 2 (in green) are composed of logos 1 and 2, respectively (Figure 2). Several reports have suggested conservation of motif1 and motif2 at the amino acid level in a variety of plants, leading to dimerization [11,27,31]. Moreover, with the help of multiple sequence alignment, we analyzed the various sequential features of 41 *Carthamus tinctorius* bHLH proteins (Figure 3). The occurrence of highly conserved motifs among same subfamily confirms the previous phylogenetic representation obtained from the proteins of bHLH domain. The corresponding position of each of these bHLH protein motifs were found conserved however, motif 1 and motif 2 was found dispersed among all subgroups. Motif 3 was present in all types of subgroups except for subfamily 7 and 8. Motif 6 was present in several proteins of subgroups 7, 8, 11, 12 and 13. The presence of motif 7 was found among the proteins of safflower bHLH from subgroup 1 and 3. Motif 8 and 9 were screened only in the proteins of 8, 9 and 10 subgroups of safflower bHLH superfamily.

Our findings revealed the presence of two major conserved domains in bHLH proteins, including the basic region with helix1 and the loop region with helix2. The results were confirmed in MEME output, which also indicated that there were two highly conserved domains (Figure 3). These results were consistent with the result of the long distance between the two motifs in CtbHLH9, as described in Figure 2. Generally, the sequences of the basic region and the position of the two helix domains were more conserved compared with the sequences obtained from the loop region. The MEME and sequence alignment results further demonstrated that our previous hidden Markov model (HMM) findings showed high reliability. The results also showed that HMM can be highly precise for predicting a given superfamily.

### 2.4. Gene Expression Analysis

Based on the RNA-seq transcriptome data (PRJNA399628; submitted on august 23rd, 2017), the RPKMs of different tissues (roots, shoots, and leaves) and flower developmental stages (F0, F2, F4, F6, F8, F10, and F12 petals) were used to examine the gene expression profiles of the bHLH superfamily in safflower. The heat map shows that these CtbHLH genes clustered into two groups. In general, genes within the same subgroup showed similar expression patterns (Appendix A). The first group was highly expressed in the petals. The second group was highly expressed in the roots. The various expression patterns indicated functional divergence of different groups of CtbHLH members in safflower.

To further study the relationship between CtbHLHs and petal development, we isolated 19 genes with RPKM ≥ 5 at F0, F2, F4, F6, F8, F10, and F12 for further qRT-PCR analysis. Of these, four genes were highly expressed at F0 (Figure 4A), and three genes were highly expressed at F2 (Figure 4B), five genes were highly expressed at F4 (Figure 4C), and four genes were highly expressed at F8 (Figure 4D). In addition, three genes were highly expressed at F12 (Figure 4E). The qRT-PCR results showed that the fold-change values of CtbHLHs were reliable.

### 2.5. Gene Ontology Annotations

The extremely deviating sequences present outside of the bHLH conserved domain suggested that bHLH proteins are actively participating in a variety of physiological processes. A large number of the *Arabidopsis* bHLH proteins were comprehensively investigated by [32,33,34,35,36,37]. Therefore, we demonstrated the biological processes associated with the safflower CtbHLH genes. For this purpose, we have used the *Arabidopsis* as the model species to perform the GO annotation of the safflower bHLH proteins. As shown in Figure 5 the majority of the safflower CtbHLH proteins were found DNA binding. Most of these CtbHLH proteins were localized in the nucleus. However, there were also CtbHLH proteins located on membrane. The biosynthetic process as well as nuclear-based metabolic consequences were linked with the majority of CtbHLH proteins. Furthermore, a moderate number of CtbHLH proteins might respond to various abiotic stresses. The phenomenon of flower development, post-embryonic development and cell differentiation involves ~25–45 CtbHLH proteins. Thus, based on these findings, the multi functional role of CtbHLH proteins could be associated with various biosynthetic and metabolic processes, in response to abiotic and biotic stresses, in the development of various tissues and organs, in signal transduction and in the cell.

### 2.6. Protein Clustering Networks

Multiple bHLH proteins can form homodimers or heterodimers that bind to DNA and regulate transcription of downstream targets; hence, protein–protein interactions have fundamental importance in bHLH protein function. Orthologous analysis of AtbHLHs was used to predict the interaction network of 41 candidate CtbHLHs. We validated 10 candidate proteins that interact with these CtbHLHs, which include a nuclear pore complex protein (1), TFs (5), E3 ubiquitin ligases (1), phytochromes (2), and repressors of jasmonic acid (1).

The independent protein interaction networks described in the Materials and methods demonstrated that the CtbHLH8 and CtbHLH16 proteins interacted with the DYT1 protein, which indicated complex regulation of certain proteins which are essential in early pollen maturation. The CtbHLH3 and CtbHLH9 proteins interact with the BTS protein and negatively regulate the response to iron deficiency and thus contributes to iron homeostasis. The CtbHLH39, CtbHLH2, and CtbHLH28 proteins interact with the TIFY10B protein and may elevate plant defense potential. The CtbHLH8 protein interacts with the AT1G59660 protein and may be involved in transport. Notably, five different proteins, including two phytochromes proteins (PHYA and PHYB) and three TF proteins (BZR1, HY5, and ABI3), were found to interact with the CtbHLH32 protein (Figure 6).

### 2.7. Topology of Conserved Cis-Elements within the Promoter Regions of CtbHLH Genes

Along with evolutionary functional divergence, the putative gene families may also exhibit diverse patterns. To identify variations in gene regulation patterns, we screened all cis-elements within the promoter regions of the selected CtbHLH genes from *Carthamus tinctorius*. PlantCARE promoter sequence analysis identified multiple cis-elements known to participate in the biological responses to stress found in the promoter region of the CtbHLH sequence (2000 bp upstream of the transcription start site, TSS), which was isolated from safflower genome database (GDB). The frequently predicted cis-elements in the promoter regions of most of the CtbHLH gene family are those associated with light response, which indicates that light signals may play an essential role in transcriptional regulation of the *Carthamus tinctorius* CtbHLH genes (Figure 7). For example, all 41 CtbHLH genes contained G-box cis-elements. The most commonly predicted cis-elements in the promoter regions of various CtbHLH genes were elements related to the light response, which indicated that light signals might play important roles in the transcriptional regulation of CtbHLH genes (Figure 7). All 41 of the CtbHLH genes contained the G-box cis-elements.

Cis-elements conserved in other major plant responses were also investigated inside the promoter regions of different CtbHLH genes, such as elements that are highly responsive to gibberellins (P-box), auxin (TGA element), jasmonic acid (CGTCA-motif), low temperatures (LTR element), salicylic acid (ARE and TCA elements), abscisic acid (ABRE element), defense and stress (TC-rich repeats), and endosperm-specific expression (RY element) (Figure 7).

### 2.8. Subcellular Localization of CmbHLH40

The hypothetical prediction of a candidate CmbHLH40 was initially estimated using the online software ProtComp version 9.0 (www.softberry.com), which was preliminarily determined to be localized to the nucleus. The transient expression assay was carried out which involved the transformation of the CmbHLH40–GFP-pCAMBIA1302 fusion construct into onion epidermal cells. It was found that the GFP signals were largely expressed inside the nucleus (Figure 8A–C). However, GFP alone exhibited a dispersed pattern throughout the whole cell. (Figure 8D–F).

## 3. Materials and Methods

### 3.1. Plant Materials

The “Jihong No.1” variety of safflower was used in this study, and the seeds were purchased from Tacheng, Xinjiang. At the end of April, 2018, the seeds were planted in the experimental field of the Engineering Research Center of Jilin Agricultural University Changchun, China, and safflower petals were collected at the flower bud stage (0 d) and 2 d, 4 d, 6 d, 8 d, 10 d, and 12 d after flowering, and quickly placed in the liquid nitrogen, wrapped in tin foil paper, and stored at −80 °C until next use. The plant expression vector pCAMBIA1302 was preserved in our laboratory.

### 3.2. Identification of Safflower bHLH Superfamily Genes

A comprehensive HMM of the bHLH domain (PF00010) was obtained from the online server of the Pfam database, which is available at http://pfam.xfam.org/ [38]. We then scanned the entire genome of safflower using the *Carthamus tinctorius* keywords by using the HMMER software tools (http://hmmer.org/) and selecting the default cut off E-value as 0.01. Assembly of the complete set of bHLH proteins sequences obtained from candidate genes along with their ID numbers were identified from the Phytozome database using the previous HMMER results (https://phytozome.jgi.doe.gov). These putative bHLH protein sequences were subsequently processed through the online CD-search tool available at https://www.ncbi.nlm.nih.gov/. The conserved domain of the *Carthamus tinctorius* bHLH proteins were further identified with the help of the online version of InterProScan (http://www.ebi.ac.uk/Tools/InterProScan/) using the following parameters: Homologous superfamilies domain and repeats, detailed signature matches, residues annotation and other features.

### 3.3. Motif Elicitation and Phylogenetic Analysis of bHLH Proteins

The conserved motifs of herbaceous *Carthamus tinctorius* bHLH protein superfamily were identified by uploading 41 amino acid sequences into the online tool of MEME web server (Version 4.8.1; http://meme.nbcr.net/meme/cgi-bin/meme.cgi). The parameters were optimized as follows: zero or one, incidence of one motif per sequence; 10 bp, motif breadth range; and three other wide ranges of motifs were identified. All different parameters follow with the default values. Multiple sequence alignment using the 41 amino acid sequences of the *Carthamus tinctorius* bHLH protein superfamily was carried out in ClustalW (Version 2) [39]. The 138 taxon members’ sequences of the *Arabidopsis* bHLH TF family were extracted from The *Arabidopsis* Info Resource (TAIR), which can be obtained from http://www.arabidopsis.org/. Visualization of the phylogenetic trees of the *Carthamus tinctorius* bHLH taxon proteins from *Arabidopsis* and *Carthamus tinctorius* were made with the ClustalW tool in combination with MEGA version 4.1 [39] using the neighbor-joining methodology, with 1000 bootstrap replicates. According to the subgroup taxonomic organization of the *Arabidopsis* bHLH proteins, a phylogenetic tree of *Carthamus tinctorius* and *Arabidopsis* was generated, following by the structural and functional classification into subfamilies.

### 3.4. Gene Ontology Annotation

GO analysis of the *Carthamus tinctorius* bHLH transporter protein superfamily was conducted using the Blast2GO tool (https://www.blast2go.com/) [40]. The full-length aa sequences of CtbHLH proteins were uploaded to Blast2Go for blast search followed by mapping and annotation. The analysis was disbursed in three categories: developmental process, molecular function, and cellular component.

### 3.5. Expression Analysis of the Carthamus bHLH Protein Superfamily

For this purpose, the safflower plants were maintained in the artificial chamber of our laboratory under controlled conditions (25C, 16h light and 8h dark) until flowering. The safflower petals frozen at −80 °C were quickly placed in liquid nitrogen and ground, and then extracted by RNA according to the manufacturer’s instructions. The extracted RNA was measured for purity and concentration of the extracted RNA by a NanoDrop 2000 ultraviolet spectrophotometer. The gene stability was detected by 1% agarose gel electrophoresis, and the RNA was stored at −80 °C. The cDNA was reverse transcribed according to the instructions of the reverse transcription kit, and the reverse-transcribed cDNA was preserved at −20 °C and then used for quantitative real-time PCR (qRT-PCR) analysis. A set of gene primers for the 41 safflower bHLH genes was synthesized based on the information obtained from the coding sequences in Primer Premier version 5 (Premier Biosoft, Palo Alto, CA, USA). To verify the integrity of primer specificity, each primer pair was generated away from the conserved domain of the genes. The internal reference genes of *Carthamus tinctorius* used in qRT-PCR analysis included 18s ribosomal RNA gene.

qRT-PCR was performed using SYBR^®^ Premix Ex Taq™ (Tli RNaseH Plus) from TaKaRa Biotechnology Co. Ltd. (Dalian, China) using the Applied Biosystems 7500 real-time PCR machine. Following the manufacturer’s instructions, a 20-μL reaction mixture was prepared with 1.0 μL cDNA template, 0.4 μM primers (F/R), 0.4 μL ROX dye, 10 μl master mix, and 7.8 μL RNase-free water. PCR conditions were set according the manufacturer’s protocol for SYBR^®^ Premix Ex Taq™. Each experiment was repeated for biological replicates of the petal samples. The relative transcript abundance values were calculated using the 2^−ΔΔ*C*t^ method [41].

### 3.6. Protein Interaction Network Prediction and Functional Annotations Using STRING

The putative *Carthamus tinctorius* CtbHLH protein sequences were added to the online server STRING version 10 (http://string-db.org). The comparative organism was specified as *Arabidopsis thaliana*. The set of genes that exhibited the highest bit scores were utilized to create the hierarchical network of proteins. The annotation information of the functional domains was pasted manually from the blast results.

### 3.7. Promoter Sequence Analysis of CtbHLH

Conserved cis-elements within the promoter region of *Carthamus tinctorius* CtbHLH genes were studied by obtaining the 2000-bp upstream flanking sequence from each putative CtbHLH TSS from the online webserver Phytozome (https://phytozome.jgi.doe.gov). Promoter sequence analysis was carried out using PlantCARE.

### 3.8. Subcellular Localization of CtbHLH40

The pCAMBIA1302 vector with the CaMV35S promoter upstream of the multiple cloning site was used to identify the localization of CtbHLH40 using the transient expression system of the onion epidermal cells. For this purpose, we isolated the *SpeI* and *NcoI* digested CtbHLH40 gene and cloned into the pEASY-T1 vector and the positive bacteria was sent for sequencing (Shanghai Biotechnology Co., Ltd. (Shanghai, China). After the sequencing, the gene fragment was digested with the aforesaid restriction enzymes and ligated into the *SpeI-NcoI* digested pCAMBIA1302 vector using T4 ligase. The recombinant plasmid pCtbHLH40–GFP was successfully transformed into onion epidermal cells, at the same time pCAMBIA1302 alone was also transformed to use as control [41].GFP expression was analyzed by scanning confocal laser microscopy.

### 3.9. Statistical Analysis

All our findings were presented as mean ± S.D. with three independent biological replicates. Differences between means of each group were assessed by one-way analysis of variance using Statistix 8.1 software. P-values equal to 0.05 was kept statistically significant.

## 4. Conclusions

Our study provides the first all-inclusive and systematic genome-wide analysis of the safflower bHLH superfamily. As a result, a set of 41 genes was characterized and classified into subfamilies. Comprehensive phylogenetics analysis of the 41 safflower bHLH proteins and 138 *Arabidopsis* bHLH proteins was carried out to identify the conserved homology between these proteins. Furthermore, protein motifs, their compositions, and their amino acid ratios within selected plant species and promoter characterization were also thoroughly investigated. Gene expression inferred by a heat map, extensive tissue-specific qRT-PCR analysis at various developmental stages, and subcellular localization of the candidate genes provided preliminary information for future investigation of the safflower bHLH gene family during flavonoid metabolism. The current investigation lays the highlights for functional characterization and confirmation of the safflower bHLH gene superfamily and improves our understanding of the bHLH gene superfamily in higher plants.

## Figures and Tables

**Figure 1 ijms-20-03044-f001:**
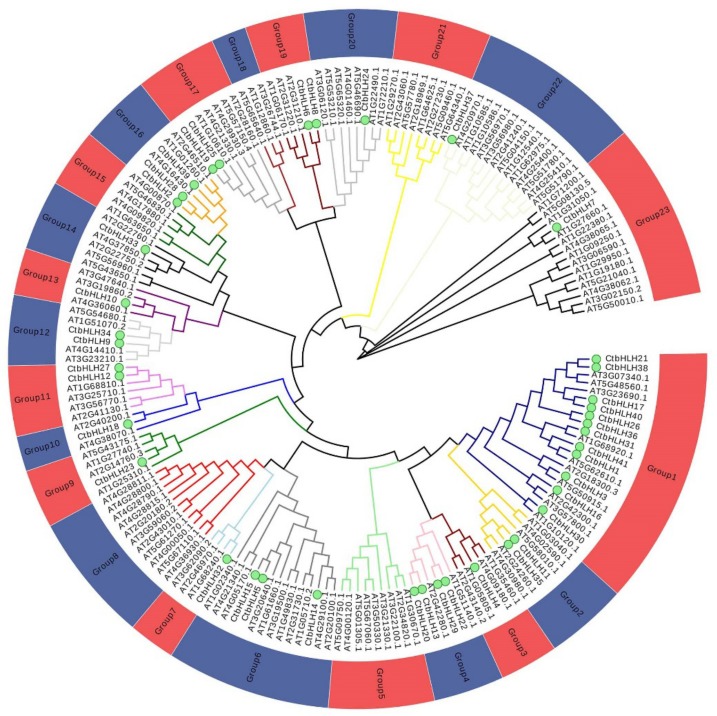
Phylogenetic tree of 179 bHLH proteins from *Carthamus tinctorius* L. and *Arabidopsis thaliana*. A neighbor-joining tree was generated using MEGA-X with 1000 bootstrap replicates. Different background colors represent different groups of safflower bHLH subfamilies. CtbHLHs are represented by green dots.

**Figure 2 ijms-20-03044-f002:**
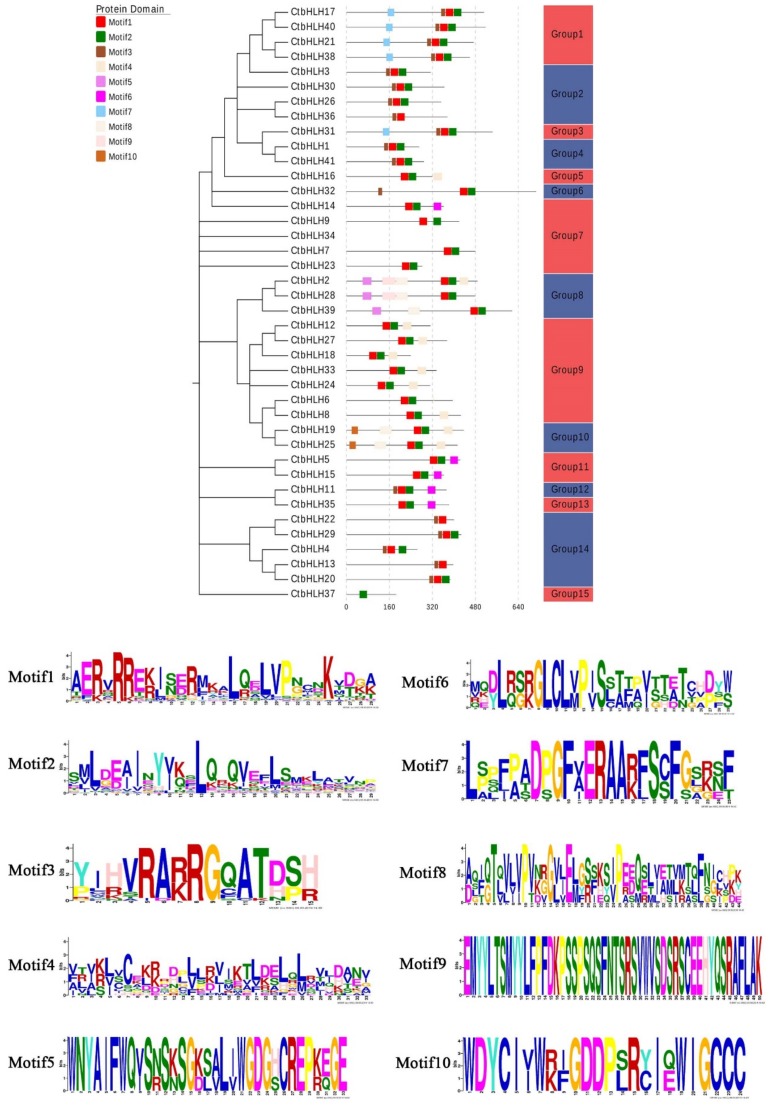
Motif distribution of safflower bHLH superfamily proteins. The phylogenetic tree was generated using the specific sequences of bHLH proteins obtained from *Carthamus tinctorius*. The online server of MEME server was used to identify the conserved motifs of *Carthamus tinctorius* bHLH superfamily genes. A total of 10 motifs were screened and visualized with different colors. The length of each grey line represents the relative length of the sequence. The occurrence of each block at different positions represents the location of the conserved motif to its matching one.

**Figure 3 ijms-20-03044-f003:**
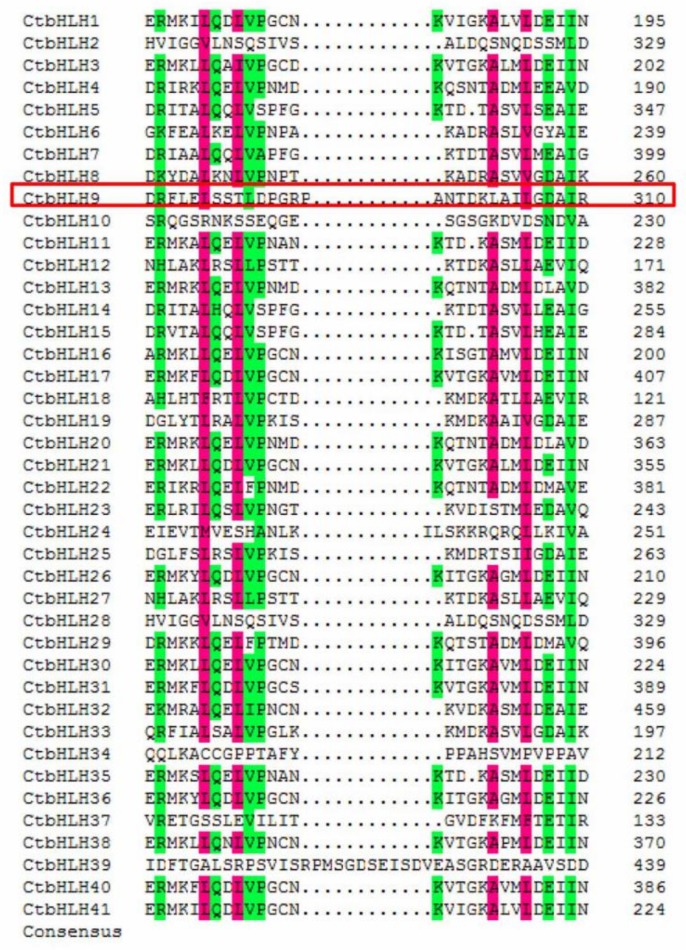
Conserved amino acid residues and pairwise alignment of the safflower bHLH proteins. The colored boxes indicate 50% identity of amino acids.

**Figure 4 ijms-20-03044-f004:**
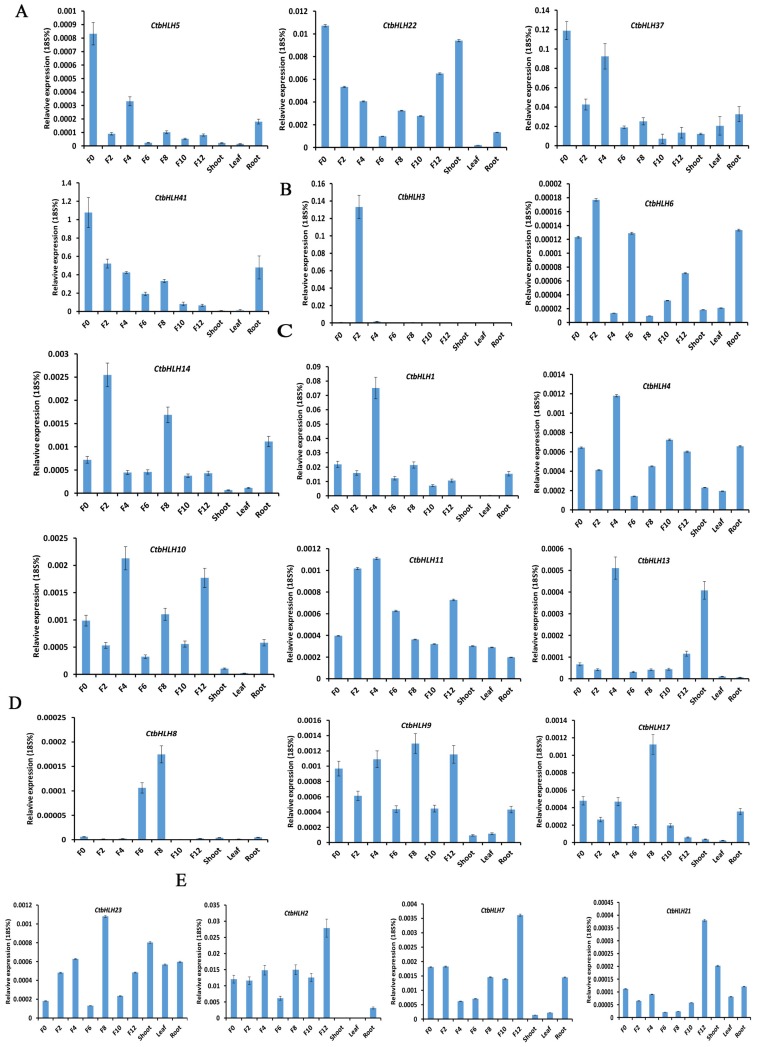
Expression profiles of 19 safflower bHLH genes during different developmental stages as determined by qRT-PCR analysis. (**A**) Includes genes that were highly expressed at F0. (**B**) Includes genes that were highly expressed at F2. (**C**) Temporal expression of genes at F4. (**D**) and (**E**) Transcript abundance of genes found at F8 and F12, respectively. Expression analysis was normalized using the 18S rRNA gene as an internal reference. Error bars were estimated from the differences in the expression patterns of three independent replicates.

**Figure 5 ijms-20-03044-f005:**
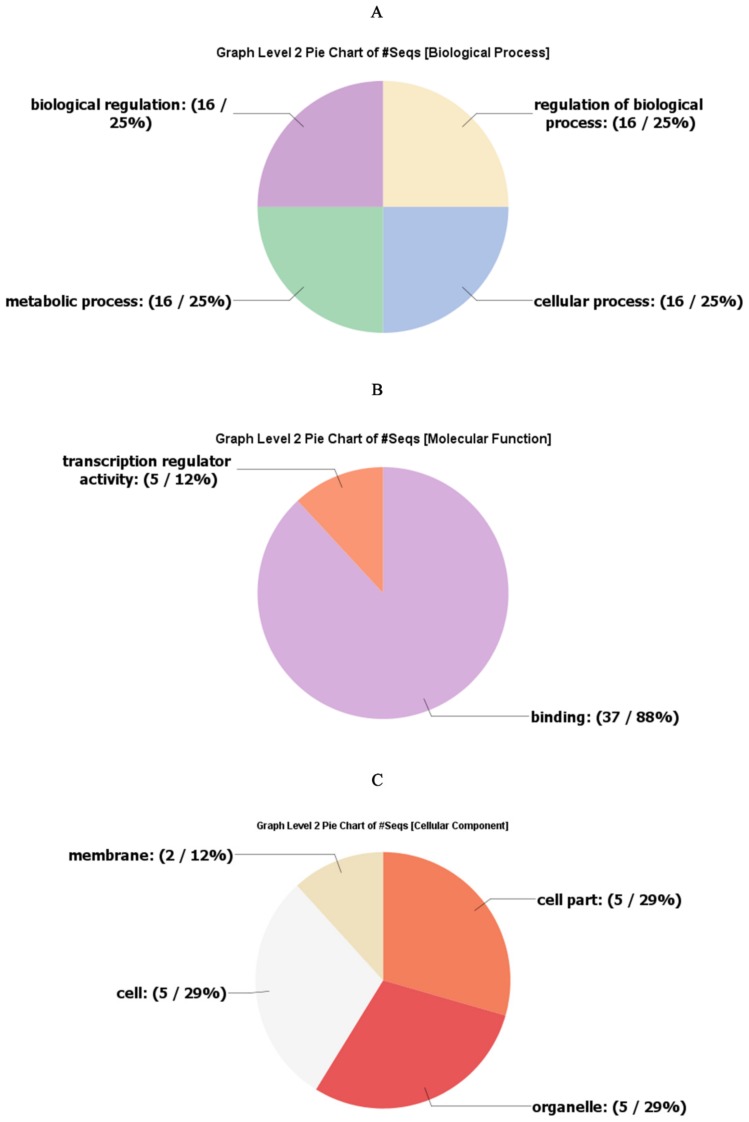
GO annotation analysis of CtbHLH transporter proteins. The annotation analysis included three major categories: (**A**) developmental process, (**B**) molecular function, and (**C**) cellular component.

**Figure 6 ijms-20-03044-f006:**
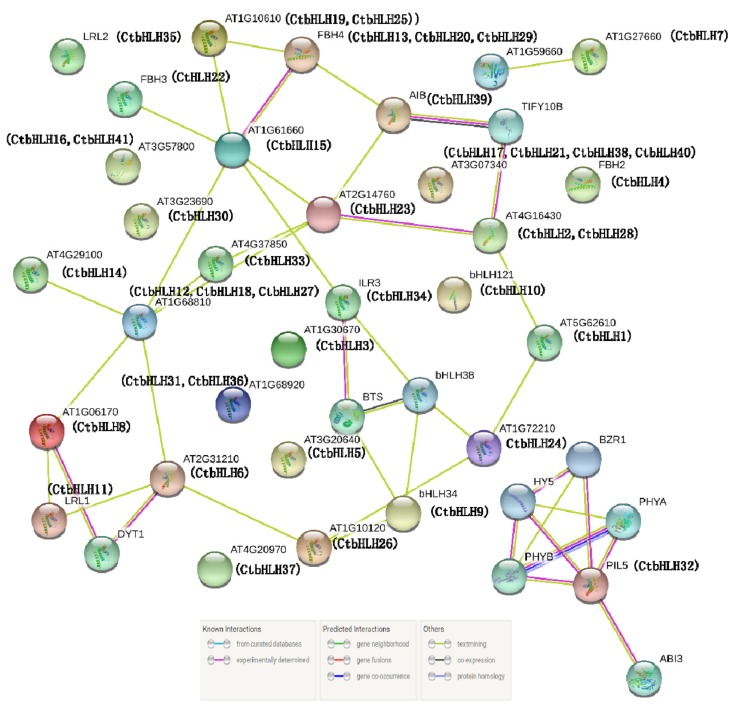
Protein association network for CtbHLHs according to information available about CtbHLH orthologs in *Arabidopsis*. The online tool STRING was used to predict the entire network. The red lines represents groups of proteins that are related to more than four other candidate bHLH proteins. The CtbHLH proteins are indicated in parentheses below the *Arabidopsis* orthologs.

**Figure 7 ijms-20-03044-f007:**
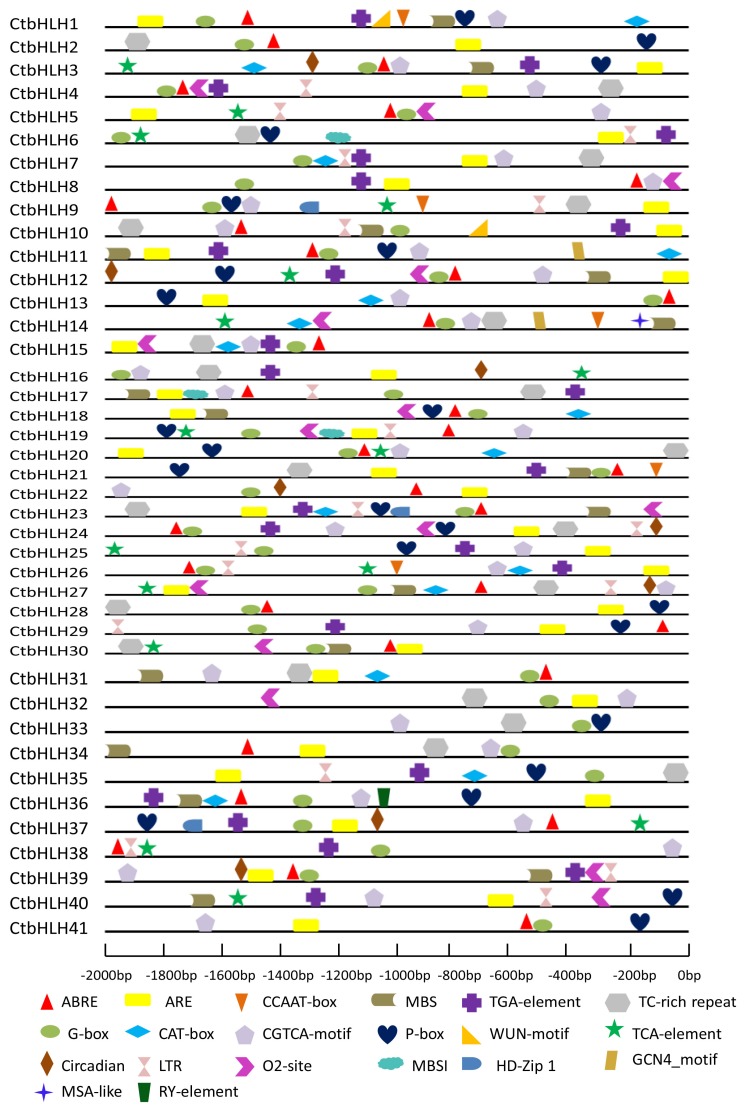
Illustration of 20 selected cis-regulatory units in all promoters of safflower bHLH genes. From the translation start site, the upstream region from −2000 to −200 shows the upstream region of each bHLH promoter. Several units, including ABRE, G-box, CGTCA-motif, P-box, and TC-rich repeats, exist in the promoter.

**Figure 8 ijms-20-03044-f008:**
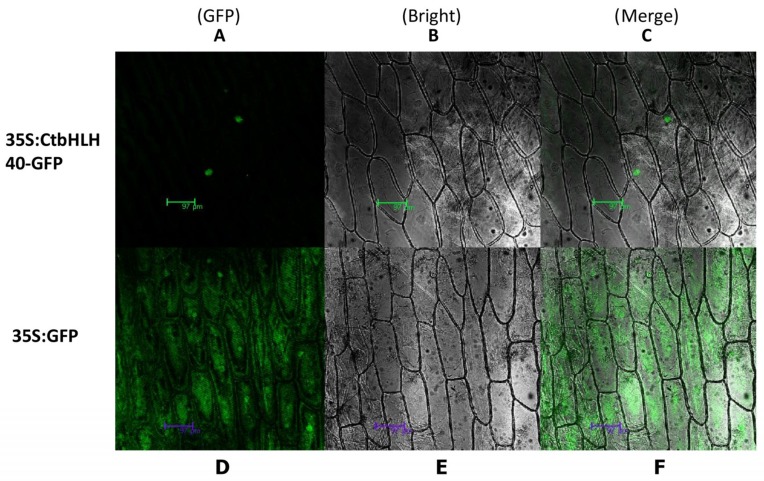
Subcellular compartment of CtbHLH40 based on transient expression in onion epidermal cells. (**A**–**C**) Onion cells introduced with 35S:CtbHLH40–GFP. (**D**–**F**) Induction of the control cells induced with 35S::GFP. (**B**–**E**) Onion epidermal cells under bright light to display the complete cell morphology. (**A**–**D**) Compilation on a dark field to visualize GFP signals. The images presented in (**C**) and (**F**) represent merging of (**A**) and (B), and (**D**) and (**E**), respectively. The flouresence was detected with the help of Leica Microsystem CMS, D-68165 Mannheim. Scar bar = 97 μm.

**Table 1 ijms-20-03044-t001:** Complete list of safflower bHLH protein family genes and their physiochemical properties. Information provided includes gene names, gene Ids, theoretical subcellular localization, coding sequence (CDS), protein length, molecular weight (MW; Da), and theoretical isoelectric points (PI). These data were obtained from ExPASy (available online: http://web.expasy.org/protparam/).

Gene Name	Gene ID	Protein Length	PI	MW (Da)	Subcellular Localization	Instability Index	*Arabidopsis* Homology	GRAVY
**CtbHLH01(4)**	CCG019410.2	271	7.84	29436.75	Nuclear	59.80	AT5G62610.1	−0.732
**CtbHLH02(8)**	CCG019793.1	488	6.85	54381.40	Nuclear	51.53	AT4G16430.1	−0.542
**CtbHLH03(2)**	CCG020831.1	314	7.78	35586.93	Nuclear	46.88	AT1G30670.1	−0.770
**CtbHLH04(14)**	CCG021763.1	264	6.66	29051.21	Nuclear	43.29	AT4G09180.1	−0.272
**CtbHLH05(11)**	CCG022142.1	424	5.63	46461.98	Nuclear	61.08	AT3G20640.1	−0.770
**CtbHLH06(9)**	CCG023511.1	396	5.54	45156.48	Nuclear	43.12	AT2G31210.1	−0.620
**CtbHLH07(7)**	CCG028074.4	480	6.73	53210.85	Nuclear	48.98	AT1G27660.1	−0.476
**CtbHLH08(9)**	CCG028576.1	426	5.62	47924.37	Nuclear	37.73	AT1G06170.1	−0.654
**CtbHLH09(7)**	CCG029033.2	420	9.34	48353.34	Nuclear	51.47	AT3G23210.1	−0.712
**CtbHLH10**	CCG029306.2	294	5.93	32779.18	Nuclear	67.22	AT3G19860.2	−1.017
**CtbHLH11(12)**	CCG031433.1	373	5.88	38966.22	Nuclear	56.54	AT2G24260.1	−0.424
**CtbHLH12(9)**	CCG031433.1	313	6.64	34357.55	Nuclear	57.98	AT1G68810.1	−0.587
**CtbHLH13(14)**	CCG003415.1	398	5.61	44597.03	Nuclear	61.12	AT2G42280.1	−0.826
**CtbHLH14(7)**	CCG003696.1	362	6.53	39563.23	Nuclear	67.71	AT1G51140.1	−0.690
**CtbHLH15(11)**	CCG004828.1	363	6.33	39459.94	Nuclear	57.93	AT1G61660.1	−0.602
**CtbHLH16(5)**	CCG006936.1	320	5.21	35518.73	Nuclear	60.39	AT3G57800.1	−0.585
**CtbHLH17(1)**	CCG009097.1	513	5.61	56099.86	Nuclear	42.63	AT3G07340.1	−0.677
**CtbHLH18(9)**	CCG010966.1	249	9.15	27766.58	Nuclear	31.23	AT1G68810.1	−0.453
**CtbHLH19(10)**	CCG015095.1	437	7.81	49777.63	Nuclear	34.05	AT1G10610.1	−0.390
**CtbHLH20(14)**	CCG016487.1	388	5.95	43102.83	Nuclear	58.48	AT2G42280.1	−0.726
**CtbHLH21(1)**	CCG030113.1	474	6.34	51930.85	Nuclear	51.00	AT3G07340.1	−0.720
**CtbHLH22(14)**	CCG000235.1	401	6.07	44439.58	Nuclear	69.25	AT1G51140.1	−0.913
**CtbHLH23(7)**	CCG000365.1	283	5.59	31696.50	Nuclear	52.80	AT2G14760.3	−0.654
**CtbHLH24(9)**	CCG001914.1	312	7.05	34294.77	Nuclear	64.73	AT1G72210.1	−0.500
**CtbHLH25(10)**	CCG002853.1	414	7.96	46510.11	Nuclear	41.81	AT1G10610.1	−0.434
**CtbHLH26(2)**	CCG003971.1	353	6.47	39567.82	Nuclear	40.90	AT1G10120.1	−0.587
**CtbHLH27(9)**	CCG006284.1	375	6.37	42123.27	Nuclear	69.08	AT1G68810.1	−0.696
**CtbHLH28(8)**	CCG006712.1	488	6.85	54381.40	Nuclear	52.40	AT4G16430.1	−0.548
**CtbHLH29(14)**	CCG008781.1	428	6.63	48008.56	Nuclear	61.50	AT2G42280.1	−0.997
**CtbHLH30(2)**	CCG009709.1	364	8.13	40547.38	Nuclear	43.90	AT3G23690.1	−0.736
**CtbHLH31(3)**	CCG010368.1	544	5.42	58619.71	Nuclear	57.98	AT1G68920.1	−0.712
**CtbHLH32(6)**	CCG011082.1	706	5.95	77127.87	Nuclear	58.02	AT2G20180.2	−0.576
**CtbHLH33(9)**	CCG011764.1	336	7.72	37338.69	Nuclear	47.03	AT4G37850.1	−0.424
**CtbHLH34(7)**	CCG013712.2	250	5.64	27865.55	Nuclear	59.41	AT5G54680.1	−0.614
**CtbHLH35(13)**	CCG014470.1	382	5.35	39191.40	Nuclear	53.49	AT4G30980.1	−0.375
**CtbHLH36(2)**	CCG017266.1	376	5.46	42211.90	Nuclear	49.28	AT1G68920.1	−0.779
**CtbHLH37(15)**	CCG018780.1	186	7.03	21187.15	Nuclear	44.33	AT4G20970.1	−0.465
**CtbHLH38(1)**	CCG018923.1	460	6.68	50715.86	Nuclear	56.68	AT3G07340.1	−0.676
**CtbHLH39(8)**	CCG018976.1	617	9.04	69705.26	Nuclear	53.97	AT2G46510.1	−0.497
**CtbHLH40(1)**	CCG019036.1	519	6.31	57019.01	Nuclear	44.67	AT3G07340.1	−0.682
**CtbHLH41(4)**	CCG019190.1	298	5.67	32673.94	Nuclear	64.80	AT3G57800.1	−0.933
**CtbHLH01(4)**	CCG019410.2	271	7.84	29436.75	Nuclear	59.80	AT5G62610.1	−0.732

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
