# Peer review of "Genome-Wide Identification, Expression Analysis, and Subcellular Localization of Carthamus tinctorius bHLH Transcription Factors"

_ijms, 2019, doi:10.3390/ijms20123044_

Round 1

Reviewer 1 Report

he manuscript “Genome-wide identification, expression analysis, and subcellular
localization of Carthamus tinctorius bHLH transcription factors” attempts to provide
functional characterization of the safflower bHLH gene superfamily that may enhance our
understanding of biochemical and physiological processes in safflower. Overall, the
manuscript conforms to the structure recommended by the journal viz. Introduction,
Materials and Methods, Results, and Conclusions. Barring few typos, the overall English
comprehension and grammar of the article is good and attains standards of a scientific article.
However, below are specific points that require improvement:
1. Below are grammatically incorrect/ambiguous sentences:
a. Line 279:281 - the sentence is grammatically incorrect/ambiguous
b. Line 276:279 - sentence is grammatically incorrect/ambiguous
c. Typo in line 139 - “wa” should be “was”
d. Line 19 - “Furthermore” needs to be replaced by “further”
2. Line 76 - please mention the keywords in the “Method” section that would be
important for reproducibility of the findings.
3. Table 1 is not of good quality and it’s an image. I suggest to replace it with text
incorporated within a table.
4. Using Motif analysis and MSA, the author identified the presence of conserved
domains across all the members of bHLH. However, it would be interesting to
know what domains are different or specific to different groups of the bHLH
proteins. Could the author comment/justify based on their analysis? Please
include that in the text.
5. Usually, the width of TFBS is considered as 10bp. In Fig 2., the author reported
TFBS with longer length (upto 50bp). Could the author perform and show motif
analysis with updated parameter settings of MEME to identify short length motifs
and see if the conclusions of the present analysis still hold true? This analysis is
important to cross check if the large motif is identified merely due to parameter
settings or are functionally relevant?
6. GO annotation (Fig 6) showed very broad terms (For e.g.: binding, membrane,
metabolic and cellular process, etc) and does not provide any relevant information
regarding the specific role of CtbHLH proteins. This analysis doesn’t make any
sense. I suggest to either remove this analysis completely or fine-tune GO analysis
to provide a specific (narrow) function.
7. Line 269 - please elaborate to be specific about “how did author optimize the
default e-value”? Also, report the final cut-off used for e-value. “e-value” should
be written as “E-value”.
8. Line 274 - what parameters did the author use for InterProScan analysis? Did the
author use the standalone or online version of the tool? Please mention explicitly
all these details as required for the reproducibility of the results in the future.
9. Line 283 - mention version of ClustalW.
10. Line 291 - I suggest to break Gene Ontology Annotation and Expression Analysis
(3.3) into two subsections.
11. Line 319-320 - “The Clustal network of protein networks” sounds ambiguous and
technical with no biological meaning.
12. Line 344-346, I don’t agree upon using “lays the foundation for functional
characterization” because the majority of the findings of the study are still based on predictions, instead of experimental validation. I would suggest using
“highlight” instead of “foundation”.

Author Response

Response to Reviewer 1 comments

Thank you very much for sending us your valuable comments on our manuscript. We have now carefully revised the overall English language of our manuscript and modified each point according to your suggestion. Please see the point by point answer to your valuable comments below.

Point 1.  Below are grammatically incorrect/ambiguous sentences:

a. Line 279:281 - the sentence is grammatically incorrect/ambiguous

b. Line 276:279 - sentence is grammatically incorrect/ambiguous

c. Typo in line 139 - “wa” should be “was”

d. Line 19 - “Furthermore” needs to be replaced by “further”

Response 1.  We feel really sorry for our carelessness in term of these grammatically incorrect sentences; according to your suggestions we have modified it now in our main manuscript file. 

a.     Please refer to (Line 294-296)

b.    Please refer to (Line 288-292)

c.      Please refer to (Line 294-296)

d.    Please refer to (Line 18)

Point 2.   Line 76 - please mention the keywords in the “Method” section that would be important for reproducibility of the findings.

Response 2. We agree with the reviewer and have added the keywords in the method section (Identification of safflower bHLH superfamily genes, please see lines 283-284) as well as added the keywords in the results section (please see line 77).

Point 3. Table 1 is not of good quality and it’s an image. I suggest to replace it with text incorporated within a table. 

Response 3. Thank you for your reminder. We have now replaced the table 1 with text inserted in a table. 

Please refer to page 3, Line 94.

Point 4. Using Motif analysis and MSA, the author identified the presence of conserved domains across all the members of bHLH. However, it would be interesting to know what domains are different or specific to different groups of the bHLH proteins. Could the author comment/justify based on their analysis? Please
include that in the text.

Response 4. We apologies and agree with the reviewer for critically reviewing our manuscript, it bring us more closer to improve the content of our manuscript. Thank you very much for the reminder. We have added the description about different domains of specific subgroups of bHLH proteins.

Please refer to Line 150-158.

Point 5. Usually, the width of TFBS is considered as 10bp. In Fig 2., the author reported TFBS with longer length (upto 50bp). Could the author perform and show motif analysis with updated parameter settings of MEME to identify short length motifs and see if the conclusions of the present analysis still hold true? This analysis is important to cross check if the large motif is identified merely due to parameter settings or are functionally relevant?

Response 5. Thank you for commenting on this, we previously performed the motif analysis of the selected bHLH super family of Carthamus tinctorius with variable lengths ranging from 2-50 bp in length in order to observe the functional overlapping of these conserved motifs and wfound no significant differences and hence suggesting the functional variance of the large motifs, however, according to your suggestion we have modified the updated TFBS width equal to 10 bp.

Please refer to Line 297.

Point 6. GO annotation (Fig 6) showed very broad terms (For e.g.: binding, membrane, metabolic and cellular process, etc) and does not provide any relevant information regarding the specific role of CtbHLH proteins. This analysis doesn’t make any sense. I suggest to either remove this analysis completely or fine-tune GO analysis to provide a specific (narrow) function.

Response 6. Thank you for your encouragement in the form of your positive comments which helped us to improve the reproducibility of our manuscript.

The Go annotation information regarding the specific role of CtbHLH proteins have been added and described in details. 

Please refer to Lines 204-217.

Point 7. Line 269 - please elaborate to be specific about “how did author optimize the default e-value”? Also, report the final cut-off used for e-value. “e-value” should be written as “E-value”.

Response 7. Thank you for your queries. We feel sorry for our typo mistake actually we want to say the selection of the default e-value. The default cut off used for E-value is 0.01. We have now replaced this error within our manuscript.

Please refer to Lines 282-284.

Point 8. Line 274 - what parameters did the author use for InterProScan analysis? Did the author use the standalone or online version of the tool? Please mention explicitly all these details as required for the reproducibility of the results in the future.

Response 8. Thank you for your encouragement in the form of your positive comments which helped us to improve the reproducibility of our manuscript.

The following parameters were used during this analysis: homologous super families domain and repeats, detailed signature matches, residues annotation and other features. We have used the online tool of InterProScan available at (http://www.ebi.ac.uk/Tools/InterProScan/.

Please refer to Lines 288-292.

Point 9. Line 283 - mention version of ClustalW.

Response 9. Thank you for your reminder. We have now added the version of ClustalW (version. 2)

Please refer to, Line 300.

Point 10.
Line 291 - I suggest to break Gene Ontology Annotation and Expression Analysis (3.3) into two subsections.

Response 10. Thank you for your suggestion. We have now split these parts of the manuscript into two sections i.e. Gene Ontology Annotation (section 3.3) and Expression analysis (section 3.4)

Please refer to, Line 309-315.

Point 11. Line 319-320 - “The Clustal network of protein networks” sounds ambiguous and technical with no biological meaning.

Response 11. We appreciate your concern about the clustering network of proteins interactions of the bHLH family. We have demonstrated the interactions network of tightly regulated proteins encoded by bHLH genes in Carthamus tinctorius. This functional web of protein–protein links extends well beyond direct physical
interactions in given specie which provides enough theoretical milestones towards understanding the comprehensive description of cellular mechanisms and functions.

Point 12. Line 344-346, I don’t agree upon using “lays the foundation for functional characterization” because the majority of the findings of the study are still based on predictions, instead of experimental validation. I would suggest using “highlight” instead of “foundation’’.

Response 12. Thank you for suggesting the correction for this sentence. We have now modified it within our manuscript.

Please refer to, Line 363-364.

Reviewer 2 Report

The study was focused on preliminary genome-wide identification, expression analysis, and subcellular localization of bHLH transcription factors within tissues of only one variety (Jihong No.1) of Carthamus tinctorius.

In the present form, the manuscript has a quite low scientific soundness and contribution to the subject. I have formulated several objections regarding the reviewed paper:

- Authors used SYBR Green fluorescent dye during gene expression studies. In this case, it is obligatory to perform Melting Curve Analysis, and results of this examination should be added in the manuscript or supplementary file (e.g., JPG or TIFF file). However, application of SYBR green dye during qRT-PCR gene quantification is not a precise method. It is strongly recommended to further re-examine cDNA samples, using gene-specific fluorescent probes (e.g., commonly used TaqMan probes).

- Tissue‐specific qRT‐PCR analysis at various developmental stages should be performed under fully controlled environmental conditions (e.g. growth chamber) or conducted under field conditions – data derived from at least three vegetative seasons. The results obtained from one year experiments are not representative and may be misleading in the context of large seasonal, phenological variability.

- Why the “Jihong No.1” variety of safflower was selected during the study? I cannot understand why only one variety was tested. In my opinion at least two-three genotypes should be investigated (in order to identify the scale of intervarietal differences)

- Description of the results and their discussion should be profoundly extended.

- Fig.5 is not readable at the present form – too many details and low graphical resolution. Error bars represent SD or SE?

Author Response

Response to Reviewer 2 comments

Thank you very much for sending us your valuable comments on our manuscript. We have now carefully revised the overall English language of our manuscript and modified each point according to your expert suggestion. Please see the point by point answer to your valuable comments below.

Comment 1. Authors used SYBR Green fluorescent dye during gene expression studies. In this case, it is obligatory to perform Melting Curve Analysis, and results of this examination should be added in the manuscript or supplementary file (e.g., JPG or TIFF file). However, application of SYBR green dye during qRT-PCR gene quantification is not a precise method. It is strongly recommended to further re-examine cDNA samples, using gene-specific fluorescent probes (e.g., commonly used TaqMan probes).

Response to comment 1.  We agree with the reviewer regarding the Melting Curve results during gene expression analysis. We apologize for not providing the data at the beginning of submission. As of now we have added the images of Melting Curve in supplementary files which were obtained during qRT-PCR expression analysis using SYBR green dye. We highly appreciate your suggestion to use gene-specific fluorescent probes such as TaqMan probes, however the transcript abundance of Carthamus tinctorius bHLH genes were found abundantly when using SYBR green dye in accordance with reference gene. That is why we considered SYBR green dye in our analysis.

Comment 2.   Tissue-specific qRT-PCR analysis at various developmental stages should be performed under fully controlled environmental conditions (e.g. growth chamber) or conducted under field conditions – data derived from at least three vegetative seasons. The results obtained from one year experiments are not representative and may be misleading in the context of large seasonal, phenological variability.

Response to comment 2. Thank you for your critical analysis regarding the expression analysis of bHLH gene family. We appreciate your valuable suggestion on the Tissue-specific qRT-PCR analysis at various developmental stages. These investigations are the part of our project and another student is currently working on it under field conditions.

Comment 3. Why the “Jihong No.1” variety of safflower was selected during the study? I cannot understand why only one variety was tested. In my opinion at least two-three genotypes should be investigated (in order to identify the scale of intervarietal differences).

Response to comment 3. Thank you for your question. The project is currently undergoing the investigations on  four different safflower cultivars, Jihong (early maturing line) (JHEM), Jihongyou sister line (JHS), Jihong No. 1 (JH1), and Jihong No. 2 (JH2) which were grown and maintained at the experimental station of the Ministry of Education Engineering Research Center of Bioreactor and Pharmaceutical Development at Jilin Agricultural University, Jilin, China. The part of this study was selected on Jihong No.1 variety on the basis of its high expression level of bHLHL genes and their correlation with flavonoid biosynthesis. 

Comment 4. Description of the results and their discussion should be profoundly extended.

Response to comment 4. We agree with the reviewer and apologies for not providing enough contents about results and discussion. We have now added the required contents throughout the results and discussion section.

Please refer to:

3.2. Identification of safflower bHLH superfamily genes

292-294

299-203

section 3.3 Motif Elicitation and Phylogenetic Analysis of bHLH Proteins

304-307

310

section 3.3. Gene Ontology Annotation

319

325. 

Comment 5. Fig.5 is not readable at the present form – too many details and low graphical resolution. Error bars represent SD or SE?

Response to comment 5. Thank you very much for highlighting the graphical resolution. We have now replaced fig.5 and added a clearer image. The error bars represented the mean value as SD.

Reviewer 3 Report

In this manuscript, Yingqi et al identify and characterize bHLH genes in the Carthanmus tinctorius genome.

This study, while similar to studies in different plant species, provides a valuable platform for future functional characterization of this class of transcription factors.

I have several minor comments:

There is no comprehensive Discussion presented.

Only 41 genes were identified in this study, which is only 1/4 of these genes presented in Arabidopsis and also considerably fewer than in peach or pea genomes. How comprehensive is this list? Is it possible that there are additional bHLH genes missed in this analysis? This is especially relevant as authors suggest that none of the genes were in subfamilies 5,8,15,18 and 21, which suggests loss during evolution, but an alternative reason might be that the list of 41 is not exhaustive.  

More specific detail would be helpful throughout. In particular, in 2.1, 41 genes were selected and identified, but what were the specific criteria (not just the list of databases used). This would help readers that are not familiar with these resources.

Table 1 is of poor resolution and should be reformatted to exclude all of the dots and arrows.

Discussion of CtbHLH36 and CtbHLH37 in Figure 2 would be interesting - since these two genes contain only one of motif 1 or 2.

The additional of the subfamilies/groups on Figure 4 might be interesting.

Figure 5 provides no additional data (just validation) than Figure 4 and could be moved to a supplementary figure.

The Gene Ontology Annotations are not well explained and I am not sure what they are adding to the manuscript. Please provide a better description of this analysis/data.

In 2.5 Protein clustering Networks, it states that authors validated 10 candidate proteins. These data should be presented in addition to the summary presented in Figure 7.

Figure 8 does not need to be divided into 3 sections.

Additional labels on Figure 9 would be helpful rather than just A,B,C,D,E,F. It is easier to read a figure when you don't have to go to the legend for the details. Specifically: GFP, dark field and merge, as well as 35S:CtbHLH40-GFP and 35S:GFP. Additionally, in the description is says GFP alone was dispersed throughout the membrane, but it looks to me to be more diffuse throughout the whole cell.

Proofreading is encouraged and small text edits are required:

-Line 45: Fe should be iron

-Line 139: motif2 "wa"

-LIne 226: define GDB

-Line 232: For example... is redundant and already stated and should be deleted

Methods for the RNA sequencing data presented in Figure 4 are not provided.

Author Response

Response to Reviewer 3 comments

Thank you very much for sending us your valuable comments on our manuscript. We have now carefully revised the overall English language of our manuscript and modified each point according to your suggestion. Please see the point by point answer to your valuable comments below.

Point 1.  There is no comprehensive Discussion presented.

Response 1.  We feel really sorry for our carelessness in term of low data on discussion section. It has been added now throughout different sections.

 Please refer to (Line 153-161) and (Lines 209-217).

Point 2.   Only 41 genes were identified in this study, which is only 1/4 of these genes presented in Arabidopsis and also considerably fewer than in peach or pea genomes. How comprehensive is this list? Is it possible that there are additional bHLH genes missed in this analysis? This is especially relevant as authors suggest that none of the genes were in subfamilies 5,8,15,18 and 21, which suggests loss during evolution, but an alternative reason might be that the list of 41 is not exhaustive. 

Response 2. We agree with the reviewer regarding the question on the number genes. We selected only 41 genes because of this data is derived from genome sequencing. The coverage of genome sequencing was 85%. A total of 89 genes were obtained by sequencing. Some of them had short sequences and no full-length sequences. Some genes have long sequences. After eliminating these invalid genes, the remaining genes were compared by BLAST, and the genes with very low homology ratio were removed. Some of them did not have the typical domain of bHLH. Finally, 41 genes were obtained.

Please refer to lines 79-83

Point 3. More specific detail would be helpful throughout. In particular, in 2.1, 41 genes were selected and identified, but what were the specific criteria (not just the list of databases used). This would help readers that are not familiar with these resources.

Response 3. We agree with the reviewer regarding the question on the information about the safflower. Based on the information available on genome sequencing, A total of 89 safflower bHLH genes were obtained by sequencing. Some of them had short sequences and no full-length sequences. Some genes have long sequences. After eliminating these invalid genes, the remaining genes were compared by BLAST, and the genes with very low homology ratio were removed. Some of them did not have the typical domain of bHLH. Finally, 41 genes were obtained.

please refer to line 79-83

Point 4. Table 1 is of poor resolution and should be reformatted to exclude all of the dots and arrows.

Response 4. We apologies and agree with the reviewer regarding the quality of the table 1 representation.  We have now formatted and replaced the table 1 with text inserted in a table. 

Please refer to page 3, Line 94.

Point 5. Discussion of CtbHLH36 and CtbHLH37 in Figure 2 would be interesting - since these two genes contain only one of motif 1 or 2.

Response 5. Thank you for commenting on this, we previously performed the motif analysis of the selected bHLH super family of Carthamus tinctorius with variable lengths in order to observe the functional overlapping of these conserved motifs. The discussion regarding CtbHLH36 and CtbHLH37 in Figure 2 have been added in the main text.

Please refer to Line 153-161.

Point 6. The additional of the subfamilies/groups on Figure 4 might be interesting.

Response 6. Thank you for your attention. Figure. 4 is now revised with the addition of subfamilies and it has been moved to supplementary file.

Please refer to supplementary files (Fig S1).

Point 7. Figure 5 provides no additional data (just validation) than Figure 4 and could be moved to a supplementary figure.

Response 7. Thank you for your suggestion. It has been modified according to your suggestion. however as per our professor suggestion, the qPCR results must be included in the paper to ensure the reproducibility of our work therefore, we have moved previous figure 4 to the supplementary files.

Please refer to supplementary files (Fig S1).

Point 8 The Gene Ontology Annotations are not well explained and I am not sure what they are adding to the manuscript. Please provide a better description of this analysis/data.

Response 8. Thank you for your encouragement in the form of your positive comments which helped us to improve the reproducibility of our manuscript.

The Go annotation information regarding the specific role of CtbHLH proteins have been added in the results and discussion section in details. 

Please refer to Lines 204-217.

Point 9. In 2.5 Protein clustering Networks, it states that authors validated 10 candidate proteins. These data should be presented in addition to the summary presented in Figure 7.

Response 9. Thank you for your reminder. We have now added the data of the 10 candidate proteins during Protein clustering Networks analysis in the supplementary files.

Please refer to supplementary file (Table. S2).

Point 10.
Figure 8 does not need to be divided into 3 sections.

Response 10. Thank you for your suggestion. we have now modified this figure into 1 section.

please to page 14 (Figure 7)

Point 11. Additional labels on Figure 9 would be helpful rather than just A,B,C,D,E,F. It is easier to read a figure when you don't have to go to the legend for the details. Specifically: GFP, dark field and merge, as well as 35S:CtbHLH40-GFP and 35S:GFP. Additionally, in the description is says GFP alone was dispersed throughout the membrane, but it looks to me to be more diffuse throughout the whole cell.

Response 11. We appreciate your valuable suggestion about adding additional labeling. We have now revised the figure 8 with a modified description of the GPF alone according to your suggestion.

Please refer to line 275.

Point 12. Proof reading is encouraged and small text edits are required:

-Line 45: Fe should be iron

-Line 139: motif2 "wa"

-Line 226: define GDB

-Line 232: For example... is redundant and already stated and should be deleted

Response 12. Thank you for suggesting the correction of these technical mistakes and correction of sentences. We have now modified it within our manuscript.

-Please refer to (Line 46)

-Please refer to (Line 146)

-Please refer to (Line 251)

-Please refer to (Line 258)

Point 11. Methods for the RNA sequencing data presented in Figure 4 are not provided.

Response 11. Thank you for your inquiry. We used Illumina HiSep 2500 to obtain RNA sequencing data and the upload date of RNA sequencing data to NCBI is 2012 under id number PRJNA399628. 

Reviewer 4 Report

In plants, the basic helix-loop-helix transcription factors play key roles in diverse biological processes. The results of this manuscript are interesting; however, I would suggest revising few points in order to improve it.

As reported by authors, safflower has a diploid genome that has not yet been sequenced; therefore there is a lack of information on and understanding of the molecular mechanisms, functional diversity, and structural genomics in this important plant.  In introduction section (line 64) were reported 41 safflower bHLH genes. This sentence is confusing. Are 41 genes the result of a comparative analyses with other plant species? Please, specify that is a result of authors analysis as reported in paragraph 1 (Genome-wide Identification and Phylogenetic Analysis of Safflower bHLH Genes) of result section.

To identify bHLH Genes in safflower, an in silico search was performed. Please clarify this aspect introducing in result section the genome database used and indicating the date of access to it.

In order to examine the gene expression profiles of the bHLH superfamily in safflower a gene expression profile was performed on the RNAseq transcriptome data. What RNAseq transcriptome data do the authors refer to? Please, clarify or add a reference.

No statistical analysis was applied for all investigates performed; please provide to do it and introduce a paragraph in the materials and methods section about statistical analyses.

The material and method section should be improved. The section is very superficial and lacks important information. The protocol of Subcellular Localization of CtbHLH40 reported is very poor and inaccurate. No references have been added. Which confocal was used?

Minor point

Please, calculate the statistical significance of differences between experimental groups in real-time PCR assy.

Please, add primers concentration, microliters shown are not useful.

Please, add the list of used primers in supplementary data.

Please, improve the quality of figures, especially that of table 1

Author Response

Response to Reviewer 4 comments

Thank you very much for sending us your valuable comments on our manuscript. We have now carefully revised the overall English language of our manuscript and modified each point according to your suggestion. Please see the point by point answer to your valuable comments below.

Comment 1.  As reported by authors, safflower has a diploid genome that has not yet been sequenced; therefore there is a lack of information on and understanding of the molecular mechanisms, functional diversity, and structural genomics in this important plant.  In introduction section (line 64) were reported 41 safflower bHLH genes. This sentence is confusing. Are 41 genes the result of a comparative analyses with other plant species? Please, specify that is a result of authors analysis as reported in paragraph 1 (Genome-wide Identification and Phylogenetic Analysis of Safflower bHLH Genes) of result section.

Response to Comment 1. Thank you very much for appreciating our research design. We feel really sorry for our carelessness. We have now rephrased (Line 64) in the introduction section.  The 41 genes of bHLH genes are selected only from the safflower genome which was comparatively analyzed with 138 bHLH of the Arabidopsis genes. This has been added in the paragraph 1 section (Genome-wide Identification and Phylogenetic Analysis of Safflower bHLH Genes)

Please refer to (Line 79-83)

Comment 2.   To identify bHLH Genes in safflower, an in silico search was performed. Please clarify this aspect introducing in result section the genome database used and indicating the date of access to it.

Response to comment 2. Thank you very much for paying attention to these details. We have now added the said modification according to the expert suggestion in the results section.

(please see line 76-78).

Point 3. In order to examine the gene expression profiles of the bHLH superfamily in safflower a gene expression profile was performed on the RNA-seq transcriptome data. What RNA-seq transcriptome data do the authors refer to? Please, clarify or add a reference.

Response 3. Thank you for your inquiry. Safflower RNA sequence was obtained by transcriptome sequencing in our laboratory and uploaded to NCBI. The sequence ID is PRJNA399628. The upload date is 2012.

Comment 4. No statistical analysis was applied for all investigates performed; please provide to do it and introduce a paragraph in the materials and methods section about statistical analyses.

Response to comment 4. We agree with the expert opinion regarding the statistical analysis during our investigations. We already added the statistics of our findings and it has been added within our manuscript. According to your suggestion we have now added a paragraph introducing the statistical analysis at the end of materials and methods section.

Please refer to Line 364-367.

Point 5. The material and method section should be improved. The section is very superficial and lacks important information. The protocol of Subcellular Localization of CtbHLH40 reported is very poor and inaccurate. No references have been added. Which confocal was used?

Response 5. Thank you for commenting on this, we are really very sorry for not providing important information when writing the material and method section which is also highlighted by other expert reviewers. we have now modified this section with details. Regarding the protocol of Subcellular Localization of CtbHLH40 reported in our study, It has also been now explained in the methodology section of subcellular localization. We have used the Leica Microsystem CMS, D-68165 Mannheim for analyzing the results.

Please refer to 

Section 3.2. Identification of safflower bHLH superfamily genes

Line 292-294

Line 299-203

Section 3.3 Motif Elicitation and Phylogenetic Analysis of bHLH Proteins

Line 304-307

Line 310

Section 3.3. Gene Ontology Annotation

Line 319

Line 325

Line 365 

Minor point

1. Please, calculate the statistical significance of differences between      experimental groups in real-time PCR assay.

2.    Please, add primers concentration, microliters shown are not useful.

3.    Please, add the list of used primers in supplementary data.

4.    Please, improve the quality of figures, especially that of table 1

Response to minor points. Thank you very much for your critical suggestions.

1.     It has been modified according to your suggestion.

2.     We have now added the primer concentrations. Line…….

3.     The list of the primers is now added with supplementary file.

4.     The table 1 is now modified to the table form. See page number 3

Round 2

Reviewer 2 Report

The Authors significantly improved the scientific level of the manuscript. In my opinion, the manuscipt may be accepted for publication after minor editorial corrections.

Author Response

We would like to thank the Reviewer for appreciating and accepting our modifications during the revision process. We have now carefully modified each point according to the editor's recommendation. Thank you once again for your nicest correspondence during the entire peer review process.

Reviewer 4 Report

I thank the authors for accepting my suggestions in order to improve the manuscript.

I would like to suggest a correction again:

Lane 183: “Based on the RNA-seq transcriptome data…”

What data do the authors refer to?

As reported in their reply, the authors should include in the manuscript the code about RNA-seq transcriptome data.

Please, report the correct date of registration data (https://www.ncbi.nlm.nih.gov/bioproject/?term=PRJNA399628)

Author Response

RESPONSE TO REVIEWER 4 (Round 2)

We are very thankful to the Reviewer once again for pointing out the following shortcomings in our revised manuscript. We have now revised these points according to your valuable suggestions. Please see the point by point answer to your comments below

Comments and Suggestions for Authors:

I thank the authors for accepting my suggestions in order to improve the manuscript.

I would like to suggest a correction again:

Lane 183: “Based on the RNA-seq transcriptome data…”

What data do the authors refer to?

As reported in their reply, the authors should include in the manuscript the code about RNA-seq transcriptome data.

Please, report the correct date of registration data (https://www.ncbi.nlm.nih.gov/bioproject/?term=PRJNA399628)

Response 1. We really apologize for our carelessness in term of these incomplete information regarding the RNA-seq transcriptome data. According to your suggestions we have now added the correct date (PRJNA399628; submitted on 23rd august 2017) of registration for this data within the main manuscript file.

Please refer to section 2.3 (Line 183)